# Hand hygiene of kindergarten children— Understanding the effect of live feedback on handwashing behaviour, self-efficacy, and motivation of young children: Protocol for a multi-arm cluster randomized controlled trial

**Glenda Dangis**[1]*, **Kirsi Terho**[1,2], **Joanna Graichen**[3], **Sebastian A. Günther**[3], **Riitta Rosio**[1], **Sanna Salanterä**[1,2], **Thorsten Staake**[3], **Carlo Stingl**[3], **Anni Pakarinen**[1]

1 Department of Nursing Science, University of Turku, Turku, Finland, 2 Turku University Hospital, Turku, Finland, 3 Information Systems and Energy Efficient Systems, University of Bamberg, Bamberg, Germany

☯ These authors contributed equally to this work.

* ghdang@utu.fi

## Abstract

Early implementation of interventions at a young age fosters behaviour changes and helps to adopt behaviours that promote health. Digital technologies may help to promote the hand hygiene behaviour of children. However, there is a lack of digital feedback interventions focusing on the hand hygiene behaviour of preschool children in childhood education and care settings. This study protocol aims to describe a study that evaluates the effectiveness of a gamified live feedback intervention and explores underlying behavioural theories in achieving better hand hygiene behaviour of preschool children in early childhood education and care settings. This study will be a four-arm cluster randomized controlled trial with three phases and a twelve-month follow-up by country stratification. The sample size is 106 children of which one cluster will have a minimum number of 40 children. During the baseline phase, all groups will have automated monitoring systems installed. In the intervention phase, the control group will have no screen activity. The intervention groups will have feedback displays during the handwashing activity. Intervention A will receive instructions, and intervention B and C groups will receive instructions and a reward. In the post-intervention phase, all the groups will have no screen activity except intervention C which will receive instructions from the screen but no reward. The outcome measures will be hand hygiene behaviour, self-efficacy, and intrinsic motivation. Outcome measures will be collected at baseline, intervention, and post-intervention phases and a 12-month follow-up. The data will be analysed with quantitative and qualitative methods. The findings of the planned study will provide whether this gamified live feedback intervention can be recommended to be used in educational settings to improve the hand hygiene behaviour of preschool children to promote health.

The trial is registered with ClinicalTrials.gov (registration number NCT05395988 https://clinicaltrials.gov/ct2/show/NCT05395988?term=NCT05395988&draw=2&rank=1).

**Data Availability Statement:** No datasets were generated or analysed during the current study. All relevant data from this study will be made available upon study completion.

**Funding:** Business Finland and the Oras Group partly funded this study. The funders did not influence the study design, data collection, data analysis, decision to publish, or manuscript preparation. No additional external funding was provided for this study. The Grant number of the award is not available. NO.

**Competing interests:** The authors declare that they have no competing interests.

# Introduction

Children's development is motivated by their efforts to understand the unknown. They learn by experimenting and doing, it is when curiosity is stimulated, that children gain knowledge [1]. From birth to 12 years old, children are active, eager to connect, and motivated social learners, making this development stage ideal for enhancing essential attitudes and behaviours [2].

Educational settings are important places to instil healthful behaviours in children and they offer environments that impact children's behaviour [3]. Early implementation of interventions at a young age fosters behaviour change and helps to adopt behaviours that promote health [4]. It is known that children's behaviour can be carried out to adulthood [1,2]. The guidance supports the development of their learning abilities, thinking, habits, and behaviours, and find their independence.

Highly infectious illnesses rapidly spread in semi-close places such as in educational settings [5]. Children cared for in day-care and preschool education have higher risks of acquiring infections than those cared for at home [6]. Children's bodies are vulnerable which make them susceptible to infections [5]. The common infectious illnesses of children are respiratory and gastrointestinal [3,5]. Illnesses that can be prevented through hand hygiene.

Hand hygiene is the most cost-effective method and fundamental behaviour to prevent the spread of infections in educational settings [5,7]. It also reduces infectious illnesses-related absences among children [7]. The commonly used method for hand hygiene promotion is through health education but promoting better hand hygiene practices is more effective [3]. Limited data are available on monitoring hand hygiene practices among children in educational settings and it is challenging because it demands time and resources.

Hand hygiene interventions are any form of techniques to be performed to promote hand hygiene behaviour [5,8]. Hand hygiene interventions that could enhance psychosocial variables such as behavioural capacity, attitudes, and subjective norms are widely used in educational settings [3,9]. Also, incorporating behaviour change techniques and play in hand hygiene interventions for children provide effective results in promoting hand hygiene behaviour [10,11] and building self-efficacy in handwashing [12,13]. Many of these studies that used hand hygiene interventions among children in educational settings evaluated effectiveness in reducing infectious related-absenteeism [5,11,14], decreasing infections [5,7], hand hygiene compliance [4,15,16] and microbiological effect [14].

Determining the mediating variables of health behaviours create a foundation for the success of the health behavioural change interventions [9,17]. One of these variables is self-efficacy, a belief in one's capabilities to accomplish a certain task and could account for the effect of an intervention [18]. It is regarded that self-efficacy is one of the variables that influence the adoption and maintenance of health behaviour [12,19]. Motivation also plays a significant role in learning and behaviour changes [8,12]. Children with a higher sense of self-efficacy tend to have better motivation [12,20].

The use of gamification as interventions have been growing due to their enjoyable elements [21]. Gamification is the use of gamified elements in non-gaming settings [22]. Studies showed that users of gamified interventions appeared to have increased motivation to perform tasks and maintain healthy behaviours [21,23,24]. Enhancing children's experiences with positive and enjoyable approaches can support them grow into healthy adults [25].

Young children's experiences with technology are becoming part of the daily context of their lives(26). Appropriate frameworks for the use of technology and interactive media are tools that can help to promote effective learning and development [26,27]. Novel digital technologies adopted in the educational environment may help to promote sustainable

handwashing behaviour of children across various settings [28]. However, current digital health technologies have been developed mainly for clinical settings [29]. This study will investigate the effect of gamified digital intervention with live feedback on preschool children aged three to six years old in early childhood education and care settings using a cluster randomized controlled trial.

## Methods

### Aim of the study

This article introduces the protocol for the Candy study: Improving children's hand hygiene which aims to evaluate the effectiveness of a gamified live feedback intervention and explore underlying behavioural theories in achieving better hand hygiene behaviour of preschool children aged three to six years old in early childhood education and care setting to promote health.

### Objective and Hypotheses

This study will evaluate the effect of the gamified live feedback intervention on promoting hand hygiene behaviour and increasing self-efficacy levels of preschool children, and if motivation crowding-out will happen. Crowding out is an effect where there is a decrease or lack of motivation to perform handwashing because the reward is already known or expected [30].

### Hypotheses

- Main effects: Control, Intervention A (instruction), Intervention B (instruction plus reward), and Intervention C (instruction plus reward and instruction in post-intervention phase) groups

H1: Feedback display will increase handwashing activity in the kindergarten (in the short term) during intervention phase

- For intervention A, B, and C groups
- For intervention B and C groups more than intervention A group

H2: Handwashing activity in the kindergartens will decline with time (while the display will still be turned on)

- For intervention A group less than intervention B and C groups during the beginning of the post-intervention phase
- For intervention B group less than intervention C group during post-intervention phase.
- For control group less than intervention A, B, and C groups in all phases.
- Motivation crowding out: Measurement-based

H3a: Effect will partially persist, when feedback display in kindergarten will be turned off during the beginning of post-intervention phase. For intervention C groups less than intervention A and B groups. (Explanation: habit will be formed, there will be partial motivation crowding out, impact in increased self-efficacy will be stronger)

H3b: Effect will vanish and handwashing will return to baseline behaviour for intervention B group in post-intervention phase. (Possible explanation: no habit will be formed, there will be no motivation crowding out)

H3c: Effect will vanish and handwashing activity will drop below baseline behaviour for intervention A group in post-intervention phase. (Possible explanation: there will be crowding out)

- Acquisition of competences and self-efficacy: Intervention A, B, and C groups

H4a: Feedback intervention will increase competences and self-efficacy level of children after the intervention phase–for intervention B and C groups stronger than for intervention A group

H4b: Feedback intervention will increase competences and self-efficacy level of children after post-intervention phase–for intervention C group stronger than for intervention B group and for intervention B group stronger than for intervention A group.

## Trial design

In this article, we describe a clinical protocol for a study using a cluster randomized controlled trial following the Standard Protocol Items: Recommended for Interventional Trials (SPIRIT) guidelines [31] (S1 Table). The study design follows the CONSORT guidelines for cluster Randomized Controlled Trials [32]. The SPIRIT schedule enrolment is reported in Fig 1.

This study will be a four-arm cluster randomized controlled trial study with three phases and a twelve-month follow-up by country stratification. This study will evaluate the effectiveness of the gamified live feedback intervention on children's handwashing behaviour, self-efficacy, and motivation in three different phases: the baseline, intervention, and post-intervention phases. In Finland, there will be a twelve-month follow-up after the baseline phase. This study will consist of four groups, the control group, and three different intervention groups. All the groups will have automated monitoring systems installed before the start of the baseline phase and will be available in all phases. The system will be de-installed after the post-intervention phase.

During the baseline phase, all the groups will have no screen activity. During the intervention phase, the control group will have no screen activity. The intervention A group will have a screen display providing handwashing instructions but no reward, and the intervention B and C groups will receive handwashing instructions and rewards from the screen display. During the post-intervention phase, all the groups will have no screen activity except the intervention C group which will receive handwashing instructions but no reward (Fig 2).

The design of this study is cluster randomized controlled, where one kindergarten serves as one cluster. Clustering will be used since the hand hygiene intervention is designed to be installed in certain areas to be used in groups. Kindergartens will be assigned as clusters rather than individuals to minimize the potential contamination among the groups, and to more accurately follow the process that will be taken at scale [33,34].

## Theoretical framework

This study will use Albert Bandura's Social Cognitive Theory (SCT) as the theoretical framework. According to this theory, the three factors: personal, environmental, and behavioural determine human behaviour and are mutually interacting among each other [18]. In this conceptual model, the human agency is functioning within the interactional causal structure to obtain and maintain a behaviour (Fig 3).

The reciprocal causation, action, cognitive, and other personal and environmental factors function as interacting determinants. This model states that individuals are not autonomous agents, but they are making a causal contribution to their motivation and action within the triadic reciprocal causation [35]. In addition, motivation is self-regulated by the influence of

| $C_0$ – Control group<br>$I_A$ – Intervention A group<br>$I_B$ – Intervention B group<br>$I_C$ – Intervention C group | STUDY PERIOD | | | | | | | | | | | | |
|---|---|---|---|---|---|---|---|---|---|---|---|---|---|
| | Enrolment | | | Baseline<br>3 weeks | | | Intervention<br>3 weeks | | | Post-<br>intervention<br>3 weeks | | | Follow-up<br>12 months<br>(Finland) |
| **ENROLMENT:** | | | | | | | | | | | | | |
| Eligibility screen | x | | | | | | | | | | | | |
| Informed consent | | x | x | | | | | | | | | | |
| Eligibility: Inclusion & Exclusion criteria | | x | | | | | | | | | | | |
| Randomisation allocation | | x | | | | | | | | | | | |
| **STUDY TRIAL:** | | | | | | | | | | | | | |
| Faucet installations - $C_0, I_A, I_B, I_C$ | | | x | | | | | | | | | | |
| No screen activity - $C_0, I_A, I_B, I_C$ | | | | x | x | x | | | | | | | |
| Education session - $C_0, I_A, I_B, I_C$ | | | | x | | | | | | | | | |
| Display installation - $I_A, I_B, I_C$ | | | | | | | x | | | | | | |
| No screen activity - $C_0$ | | | | x | x | x | x | x | x | x | x | x | |
| Display with instructions - $I_A$ | | | | | | | x | x | x | | | | |
| Display with reward - $I_B, I_C$ | | | | | | | x | x | x | | | | |
| Deactivation of displays - $C_0, I_A, I_B$ | | | | | | | | | | x | | | |
| No screen activity - $C_0, I_A, I_B$, | | | | | | | | | | x | x | x | |
| Display with instructions but no reward - $I_C$ | | | | | | | | | | x | x | x | |
| Deinstallation of faucets - $C_0, I_A, I_B, I_C$ | | | | | | | | | | | | | x |
| **ASSESSMENTS:** | | | | | | | | | | | | | |
| Automated monitoring system - $C_0, I_A, I_B, I_C$ | | | | | | | | | | | | | |
| - Time and soap usage | | | | x | x | x | x | x | x | x | x | x | |
| Children interview - $C_0, I_A, I_B, I_C$ | | | | | | | | | | | | | |
| - Age, gender, self-efficacy, and motivation | | | | | x | | | x | | | x | | |
| Children observation - $C_0, I_A, I_B, I_C$ | | | | | | | | | | | | | |
| - Age, gender and hand hygiene behaviour | | | | | x | | | x | | | x | | |
| Parents' survey - $C_0, I_A, I_B, I_C$ | | | | | | | | | | | | | |
| - Children's age, gender and observed hand hygiene behaviour at home | | | | | | | | x | | | | | |
| Kindergarten staff interview - $C_0, I_A, I_B, I_C$ | | | | | | | | | | | | | |
| - Children's observed hand hygiene behaviours in kindergartens | | | | | | | | | | x | x | | |
| Children interview - $C_0, I_A, I_B, I_C$ | | | | | | | | | | | | | |
| - Age, gender, Self-efficacy, and Motivation | | | | | | | | | | | | | x |
| Number of sick leave days - $C_0, I_A, I_B, I_C$ (Finland) | | | | | | | | | | | | | x |

**Fig 1. Schedule of enrolment, intervention, and assessments.**

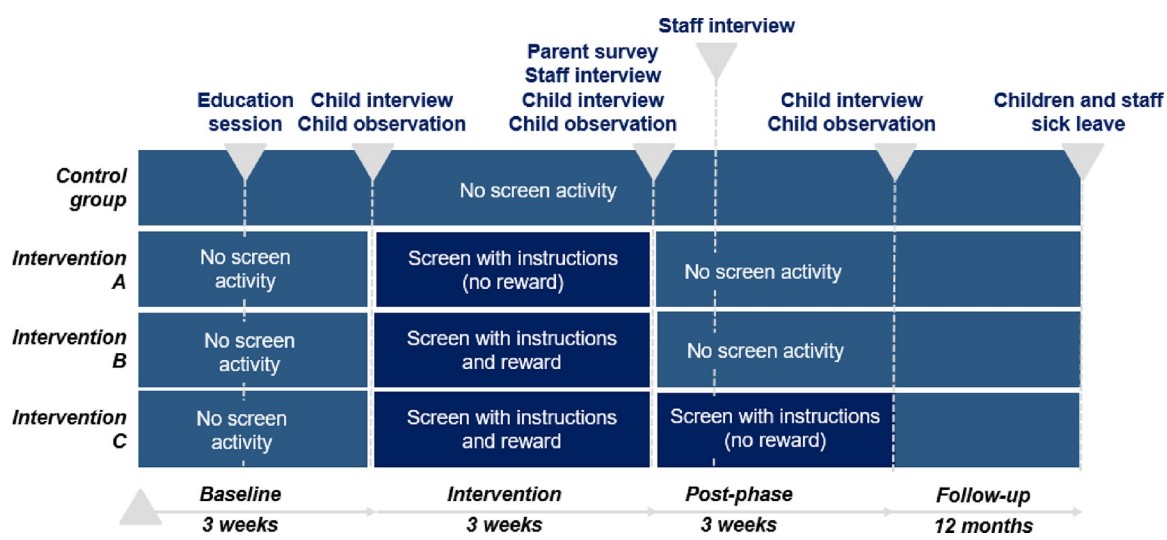

**Fig 2. The design and data collection of the study.**

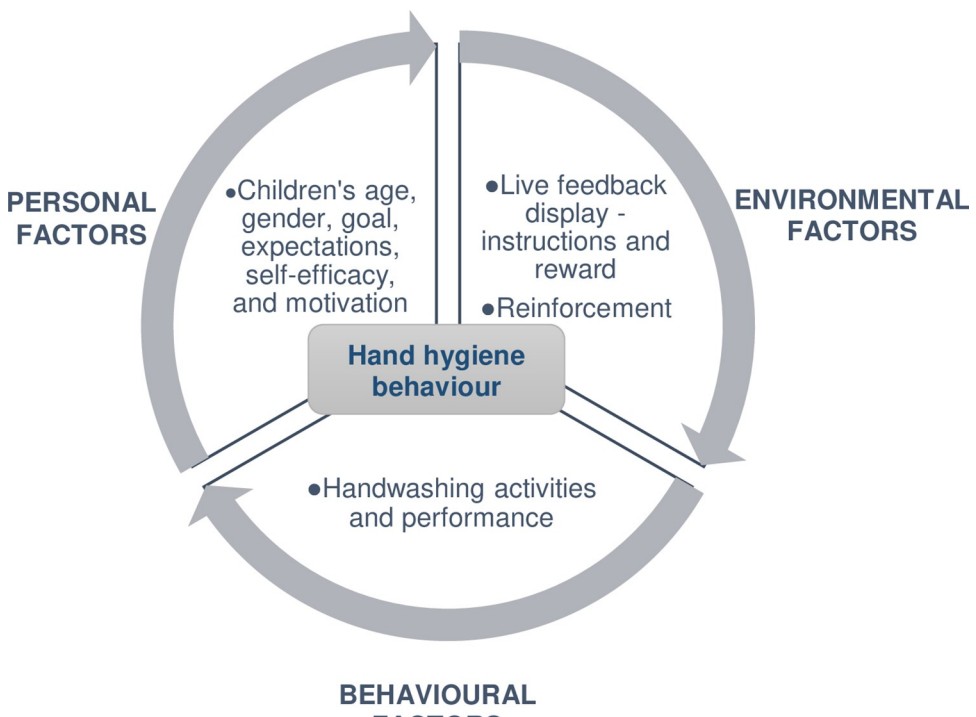

**Fig 3. Illustration of the reciprocity of the factors determining hand hygiene behavior.**

proactive and feedback mechanisms, but the adaptation of humans will depend on the fore-thought of feedback control of action [18].

The technology to be used in this study that provides live feedback (instructions and reward) when performing handwashing can build positive outcome expectations, enhance self-efficacy, and attain a goal (Fig 3). This kind of intervention can promote to maintain and improve hand hygiene behaviour by enhancing motivation and self-efficacy. Figs 3 and 4 show how the theory is applied using the live feedback intervention.

## Study setting

This study will be conducted in kindergartens of different locations around Turku, Finland and Bamberg, Germany. In Germany and in Finland, kindergartens are under the supervision

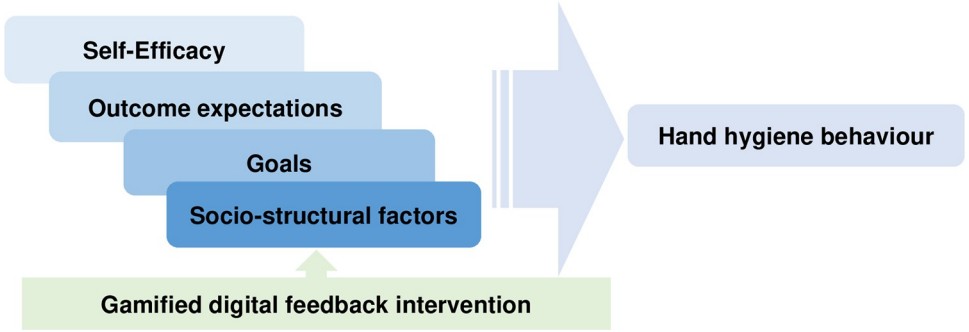

**Fig 4. The application of the SCT using the live feedback intervention.**

of Early Childhood Education and Care (ECEC) [36]. It is a part of Finnish and German education systems which support the children's development and focus on learning through play. It is an essential stage on the child's path of growing and learning. ECEC is a systematic and goal-oriented entity that includes upbringing, education, and care, specifically emphasizing pedagogy aiming to promote and support holistic growth, development, health, and wellbeing (Act on Early Childhood and Care). Groups of children in the kindergartens may be formed in various ways, considering the children's ages or needs for support. Distribution on staffing and maximum group sizes are considered in forming the groups of children. The kindergarten staff are composed of preschool teachers and assistants who are responsible for the daily activities of the children [36–39].

## Eligibility criteria

The kindergartens around Turku, Finland and Bamberg, Germany will be eligible to participate in the study. The criteria for the participants will be A) children enrolled in the kindergartens aged two years and six months to six years old, their parents or legal guardians, and kindergarten staff who will provide observational measures; B) give informed consent by both the child (verbal consent) and parents/legal guardians and kindergarten staff; C) can understand and communicate Finnish or German. The exclusion criteria will be participants that cannot answer the assessment data.

## Outcome measures

The primary outcome is the hand hygiene behaviour of the children. The system will measure the hand hygiene activities which includes the start time of extraction, water volume, water temperature, and end time of extraction; from these data, the following data will be derived: average flow rates, approximation of energy consumption, quality and duration of handwashing, and time stamp of soap used.

The secondary outcomes are self-efficacy, intrinsic motivation, hand hygiene behaviour at home from parents' survey, hand hygiene behaviour in kindergartens from kindergarten staff, and hand hygiene behaviour in the kindergarten by observation sessions. Self-efficacy and intrinsic motivation will be measured from the children's one-on-one interview, a three-point Likert scale with open-ended questions. The children will be asked easy and simple questions about their hand hygiene performance, self-regulation on motivation, emotional states, and decisional in performing handwashing. They will respond by pointing their answer from the three smiley faces (sad, neutral, and happy). Colourful smiley faces will be constructed to catch the attention of the children. Each question is followed by open-ended questions to explore their opinions regarding the topic asked and to support their answers. The children interview includes introduction of the interviewer and asking simple questions regarding their day or regular activities for the children to feel comfortable. The parent survey will be an electronic survey composed of five-point Likert and open-ended questions about the observed hand hygiene behaviour of the children at home. The kindergarten staff interview will also include five-point Likert and open-ended questions, evaluating the hand hygiene behaviour of the children in the kindergartens. The assistant researchers will observe the hand hygiene behaviour of the children during the handwashing activity in the kindergartens. Observation sessions will include a checklist consisting of the phases of handwashing, the checklist will follow the hand hygiene guideline of the United Nations Children Fund (UNICEF).

The tertiary outcome is the number of sick leave days of the children and kindergarten staff (applies in Finland) considering the seasonal change.

## Study participants

The population of the study will be the preschool children who are enrolled in kindergartens. The children aged between 2.6 to 6 years old, their parents, or legal guardians, and the kindergarten staff who will be providing the observational measures.

## Sample size

The sample size is calculated based on the significant effects found during the pre-study. In the pre-study, 10,000 handwashing events were recorded over a period of 42 days in a treatment group (n = 82 children in two clusters) and a control group (n = 53 children in two clusters). Using Cohen's power analysis with $f^2$ = 0,35 (large effect size) and power level of 0.8 and 2 predictors and a probability level of 0.05 leads to n = 31. However, since randomization is on a cluster level, the n must be increased. With the ICC from the pre-study of 0.062, and estimated cluster size (size of each kindergarten) of 40 children, we calculate the design effect:

$$D = 1 + (m - 1) * ICC = 1 + (40 - 1) * 0.062 = 3.418$$

Thus, we need to increase the sample size by 3.418 compared to individual randomization, due to cluster randomization leading to 106 participants (n = 31 * 4.418 = 106). With the assumption of a cluster size of 40 children per kindergarten, this leads to three kindergartens that need to participate in the study. Each kindergarten represents one cluster and gets assigned to be the control, the intervention A, B, and C groups. Two intervention groups will be covered by one cluster.

## Recruitment

The recruitment process was started through the kindergartens' software platforms, email, presenting a short demo video on the platforms regarding the study, and information leaflets. A website link of the study's information, recruitment, and registration was provided in the medium used in the kindergartens. In Finland, the registration will be through a webropol link and emails. In Germany, the registration will be through emails. The anonymity of the participants from the research team will be secured during the process.

## Randomization

The cluster randomization will be at the kindergarten level. Kindergartens will be randomly assigned to control or intervention groups by two researchers that are not members of the research team and are stratified by country. One kindergarten serves as one cluster. Simple randomization will be used which will be drawing lots, where each cluster will be assigned a number in a piece of paper. The parents will enrol their children for participation in the study within the time set through the website link provided and email.

No information will be given regarding the allocation. Allocation will be about four weeks before the start of the baseline phase and will be conducted by the two non-member researchers. They will generate the allocation sequence, enroll the participants, and assign them to control or intervention groups. There will be complete concealment of the allocation from the members of the research team and the participants, to exclude the possibility of predicting the next allocation [33].

## Blinding

Due to the intervention's design and purpose, the research team and the participants will have the possibility to know the assignment. Thereby, this study will use double-blinding in which

the children, parents, and kindergarten staff, and the statistician will be unaware of which group will receive the content of the intervention. In the informational letters, the feedback display will not be mentioned, the children and kindergarten staff will only know during the intervention phase. Assistant researchers will also be hired to interact with the participants. They will be given limited information, will be unaware of the study's design, and which group received the content of the intervention.

## Intervention development

The technology for the gamified digital feedback intervention was developed in Germany and the content was developed by the whole research team. It is a child-centred smart hand hygiene intervention. The installations at each washbasin are IoT-based. Each installation will consist of a touchless faucet powered by a digital power box (a), a soap sensor (b), a feedback display (only for treatment groups) (c), and a gateway (d), which connects to the devices (a-c) via Bluetooth and relays data to a cloud infrastructure (e). Connected Information Systems (f), such as dashboards to monitor hand hygiene, can access the system via this cloud infrastructure [40] (Fig 5).

The digital power box is a device developed by Amphiro AG that equips faucets with digital functionalities. The device will be installed between the water wall outlet and the faucet in the kindergartens. It measures, processes, and sends the individual water withdrawal data to an LTE Gateway. The energy required for its operation will be generated by an integrated micro-turbine, which eliminates the need for an external power supply (often unavailable at the installation site) or batteries (which need to be replaced regularly and are therefore impractical for many maintenance processes).

This study will use a well-tested technology. It is currently used in clinical settings to improve the hand hygiene of health care professionals. The technology was tested during a pre-study conducted in spring of 2021. This kind of similar live feedback approach used in the field studies has shown larger effects on routine behaviour [41]. The screen display will provide an educational video showing instructions on how to do handwashing when hands are dirty

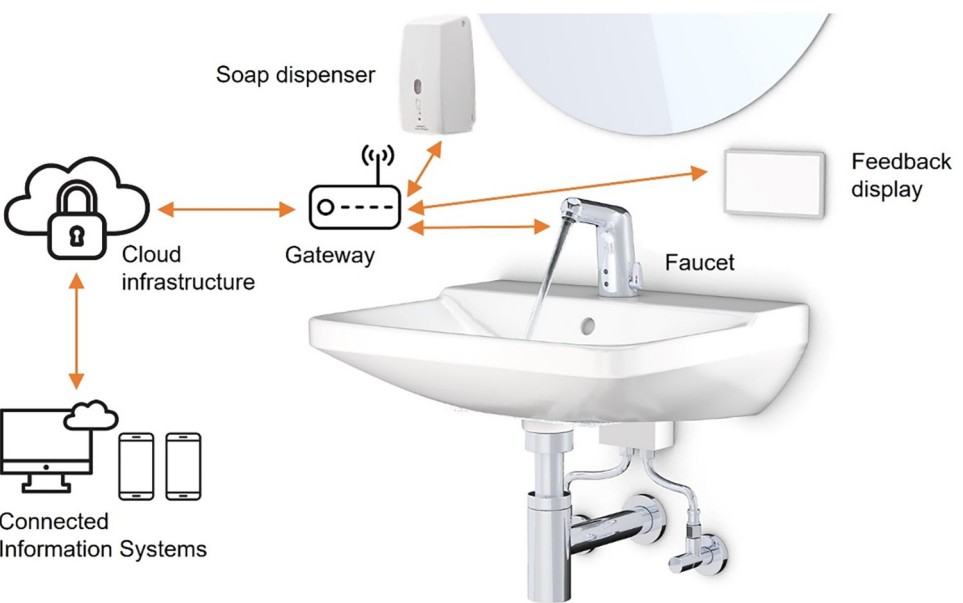

**Fig 5. Technical setup of the system.**

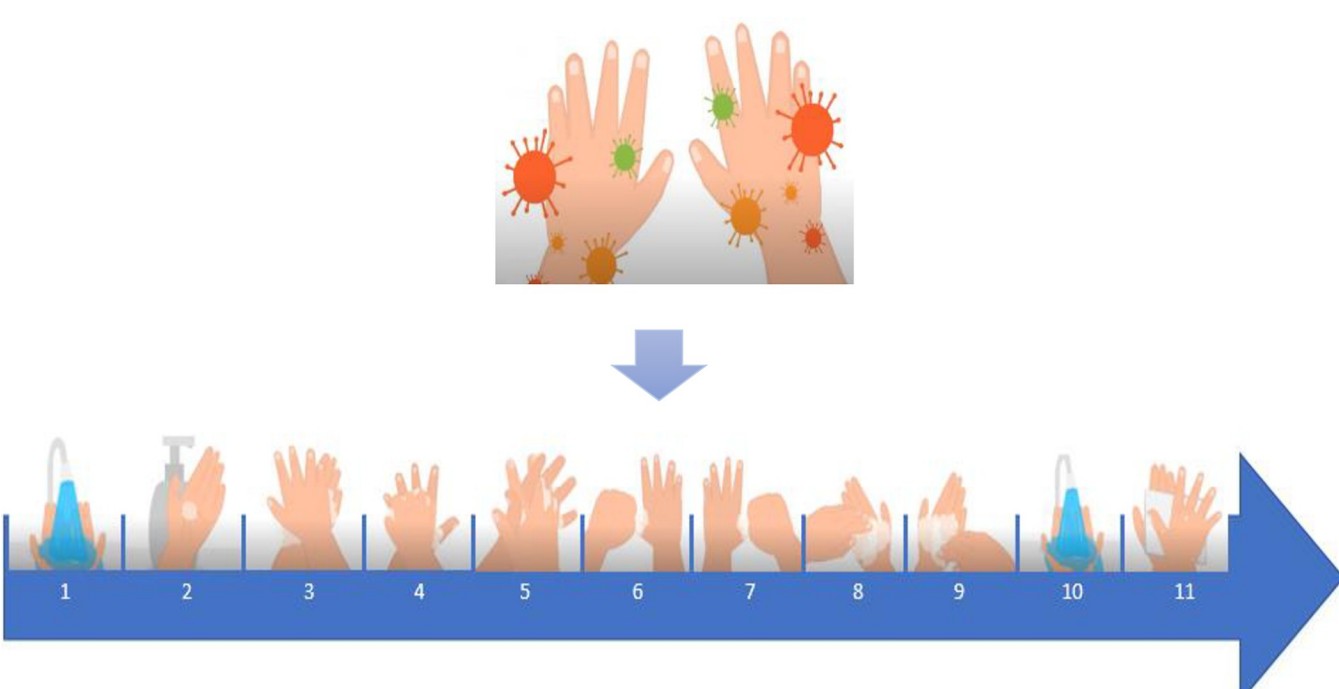

**Fig 6. The "How to handwash" content of the video on the screen display.**

for the children to follow (Fig 6). The handwashing instruction was guided by the World Health Organization's "How to Handwash" poster. When the handwashing is done correctly (based on the water and soap usage), a reward of animal animation will appear on the screen display (Fig 7). The live feedback display will enhance the children to do handwashing correctly, thus improving hand hygiene practices.

## Implementation

This study will be composed of three phases, the baseline, intervention, post-intervention, and a twelve-month follow-up (Fig 2). The first phase will be the baseline which will last for three weeks. The systems will be installed in the kindergarten sinks which will collect data from the handwashing activities, but it will not provide any feedback or screen activity. A "how-to" teaching session (video-based training) will be held in each kindergarten in the middle of the baseline phase for all children to learn the same hand hygiene information. The second phase will be the treatment and will last for three weeks. The third phase will be the post-intervention, and it will also last for three weeks. After this phase, the systems will be de-installed from all the kindergarten sinks. In Finland, there will be a twelve-month follow-up after the intervention.

The control group will have no screen display next to the sink throughout the study. The intervention A (instruction) group will receive instructions from the screen display next to each sink where handwashing instructions are shown during the handwashing activity but no reward after. The intervention B (reward) and intervention C (reward plus instruction) groups will receive instructions and rewards shown on a screen display right next to the sink during the intervention phase. In these groups, handwashing instruction is shown during the handwashing activity and if handwashing is done correctly (based on time and soap usage), a reward of animal animation is shown on the screen, see Figs 6 and 7. During the post-

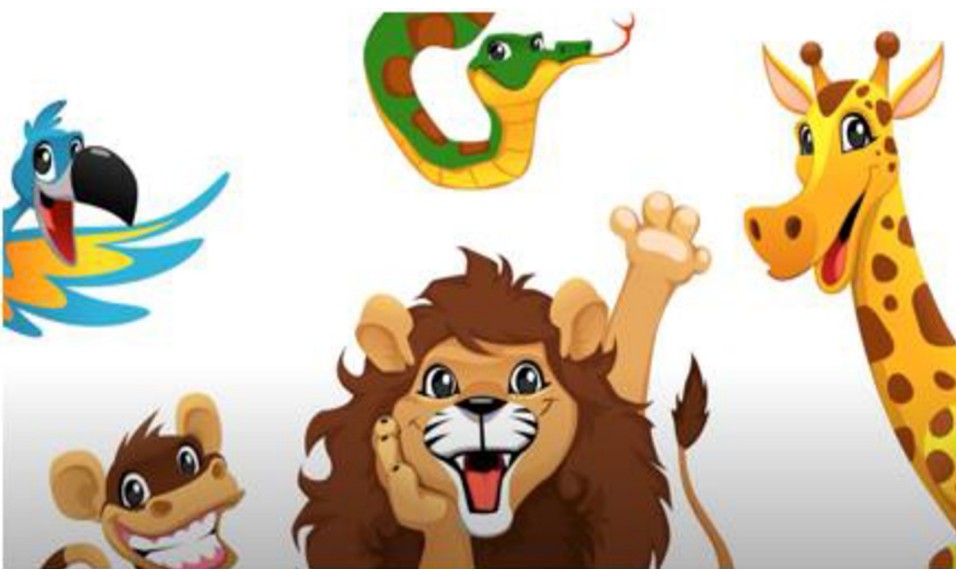

**Fig 7. The animal animation as a reward.**

intervention phase, the intervention C group will remain to have the screen display receiving instructions with no reward. In Finland, sick leave days of the children and kindergarten staff will be tracked (Fig 8).

To improve the adherence of the participants to the study, the research team will maintain the engagement, encourage retention, and thank the kindergarten managers, staff, children, and parents. Named persons from the researcher team call and pay visits to the study settings every now and then if the kindergarten personnel permits. Contact information of the research team will be provided to the kindergarten staff and parents before the trial. The participants will be reminded through email about the oncoming surveys and interviews, and they will be updated about the progress of the study.

## Data collection

This study will start in mid-year of 2022 and will last for a year. The data will be analysed using quantitative and qualitative methods. The child's age and gender will be collected. The implementation phases and timeline of the data collection are shown in Figs 1 and 2.

The handwashing activity data from the system will be recorded every day during the baseline, intervention, and post-intervention phases. The parent survey data will be given and collected within the last week of the intervention phase. The kindergarten staff interview data will be collected during the last week of the intervention phase and right in the beginning week of the post-intervention phase to evaluate the observed effect of children's hand hygiene behaviour and reactions towards the removal of the instructions and reward. The children's observation sessions and interviews will be collected during the last weeks of the baseline, intervention, and post-intervention phases. The number of sick leave days will be collected from the sick leave records provided by the kindergarten staff one year before and after the start of the baseline phase.

The follow-up strategy will include sending updates of the study phases through email to the kindergarten managers and staff. The email includes the details of reminding how and when the participants will be contacted for follow-up. It will also include updates on the contact information of the participants.

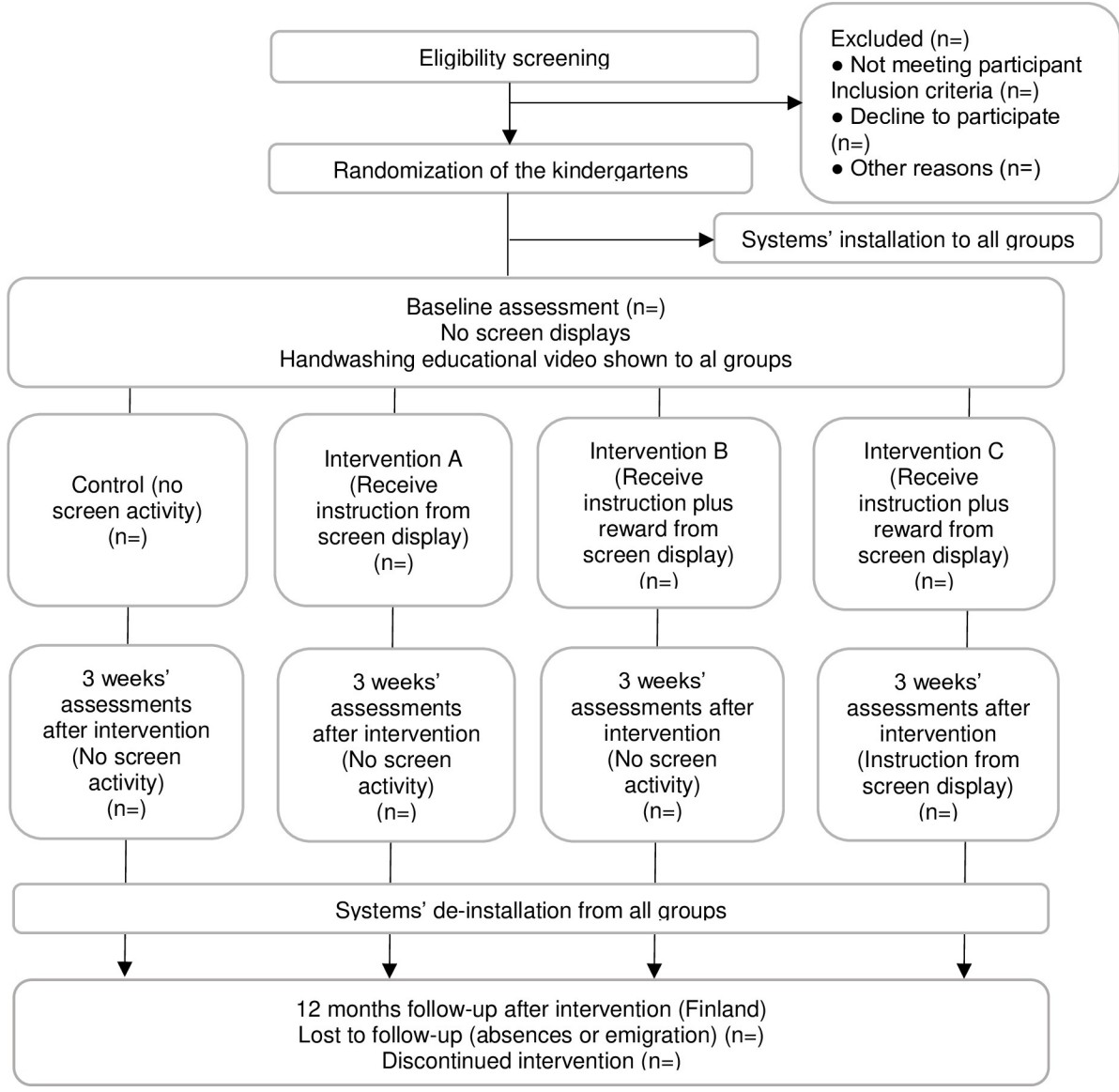

**Fig 8. Flowchart of the cluster randomization.**

## Data analyses

Data will be analysed using qualitative and statistical methods. All eligible participants will be included in the analysis. The statistical analyses will be performed with R. The categorical and numerical data will be analysed using one-way analysis of variance (ANOVA). Qualitative analysis will be presented thematically [42].

- For the measurement data from the system:

Data pre-processing: Data will be investigated where water and soaping data are combined to single handwashing procedures. Data points that are not valid (outside of kindergarten opening hours) will be removed. The descriptive statistics of the clusters, randomization checks, ICC (intracluster correlation) will be calculated. The data will be analyzed using linear

fixed-effect regression model for panel data, and dependent variable soaping time. The four groups will be compared in baseline, intervention, and post-intervention phases.

• For the survey and staff interview data:

The comparison among the groups will be implemented using random effects regression, allowing for the clustered nature of the data, and adjusted using the covariates of the adjusted analyses (when a small number of clusters are present) [43]. Simple bias analysis will be used to quantify the uncertainty of the biases such as confounding variables [33]. The qualitative results of the survey and kindergarten staff interview will be analysed and compared thematically. The four groups will be compared after intervention phase.

• For the children interview and observation data:

The analyses will be compared among groups based on the complete data only. The comparative analyses approach used to calculate the observation, survey, and staff interview will be used. The self-efficacy measurement scale will be developed and evaluated for its psychometric properties [44,45]. The qualitative result of the observation and interviews will be analysed and compared thematically [42]. Children who will participate the interview at one time point will be considered as missing data, these will be excluded from the analysis. The four groups will be compared in baseline, intervention, and post-intervention phases.

For the sickness day data:

The total sick leave days are compared among the groups one year before and after the start of the baseline phase.

The statisticians of the two universities will validate the statistical results of the data.

## Data management

The anonymity of the participants using the faucets will be preserved as no person-specific data is collected. The experimental setup and data collection will not allow the collection of personal data and draw conclusions about the individuals. However, since this study will be a cluster randomized trial, the data collected will be the summary data for each arm. In addition, consent will be asked for the collection and storing of the observation, survey, interview, and sick leave data from the participating children, parents, and kindergarten staff. The behavioural data of the experiments will be stored in associated databases. All the data CSV files will be used anonymously. The data collected will be stored in both universities.

Research results will only be published in an anonymized form that does not allow conclusions to be drawn about each child. To ensure that behavioural data is protected from being captured by a third party, access to all critical infrastructure services will require authentication and authorization. In case a third party steals a faucet, the faucet's access to the server will be remotely disabled. The documentation and the software components will be stored in git-repositories which have a version control functionality. The data communication and data storage will comply with GDPR accordingly.

## Data monitoring

The data will be handled in a safe environment and will not be distributed among groups. This behavioural intervention poses a very low risk, it does not require trial steering or data monitoring committee [46].

## Auditing

No auditing will be planned.

### Access to data

Only authorized members of the research team will have access to the collected data.

### Ethical considerations

This study Candy–Hand hygiene for children, full-scale study was reviewed and granted ethical approval by the Health Care Division of the Ethics Committee for Human Sciences of the University of Turku (S1 File) in November 2021 and Ethics Committee of the University of Bamberg (S2 File) in January 2022. This study will comply with the ethical principles for research with human participants involving minors provided by the German Research Foundation [47] and Finnish National Board on Research Integrity [48]. This study will involve minors, and will strictly follow the ethical research involving children [49], which have the right to be informed about the study in a way that they can understand and give their consent to participate in the study. The participants will give their informed consent to voluntarily participate or refuse to participate and to withdraw at any time during the study without any negative consequences. The participants will receive information letter about the contents and truthful aims of the study and any potential harms and risks. Inform consent and information letters will be in electronic and paper forms. The children will be informed about the study in an understandable manner and will give their consent verbally to participate in the study. The participation of the children will be primarily decided by the parent or legal guardian. The researchers will always respect the autonomy and voluntary participation of the participants. There will be no personal data collected during the study, the identity of the participants will remain anonymous. The data collected will be used and kept confidentially for research purposes only. The data will be stored for at least 10 years after the study has been completed.

**Dissemination and protocol amendments.** The results from this study will be published in a peer-reviewed journal irrespective of the study results. Authorship of publications will be granted according to the criteria of the International Committee of Medical Editors. We plan to make the anonymized data set, including the data from the published articles, available as a supplementary file of the main publication.

## Discussion

The gamified live feedback intervention aims to improve the hand hygiene practices of preschool children in kindergartens. By giving real-time feedback (handwashing instructions and rewards) during the handwashing activity, the self-efficacy and motivation of the children are enhanced, attitude and confidence are boosted which leads to improvement of hand hygiene behaviour. A promising hand hygiene behaviour change intervention should include "demonstration of behaviour", "instruction on how to perform the behaviour", and "adding objects to the environment" such as installing handwashing stations and providing soap [10]. It is important for the children to understand how to perform handwashing in a correct way and provide the right tools to be able to do so [10]. Digital technologies such as computer games, videos, and video cameras have shown to improve and monitor handwashing behaviour, self-efficacy, and hygiene knowledge among young children in educational settings over a short period of time [28]. However, combinations of sensor and digital feedback technology have not been used in the studies. Another challenge is the lack of behavioural theories used in previous studies in the design stage. To our knowledge, this study will be the first to implement a gamified live feedback display focusing on improving the hand hygiene behaviour of children aged three to six years old in early childhood education and care setting.

This study will use multiple measurements such as a monitoring system, survey, interview, and observation to assess hand hygiene behaviour, self-efficacy, and motivation. This will

provide versatile data to support the effectiveness of the intervention. It will also provide information if self-efficacy and motivation play important roles in attaining hand hygiene behaviour. Another is to provide further evidence if preschool children's self-efficacy is a significant variable in determining the effectiveness of the intervention and obtaining hand hygiene behaviour. In this study, hand hygiene self-efficacy for preschool children will be developed and evaluated which up to these days none have been developed. This study will also measure the sick leave days of the children and kindergarten staff which could give information if using the intervention is effective in reducing the number of sick leave days. Hand hygiene interventions in educational settings showed to decrease the spread of infectious illnesses [3,7] and reduce absenteeism [3]. This study will use a cluster randomized controlled design which includes larger participants and a long follow-up period. Systems will be installed in all the groups which can provide concrete data on the handwashing activities. Another strength of this study will be assessing the children's self-reports as there are few interventional studies that measured this. Evaluating the effectiveness of the gamified live feedback intervention will enable the research team to determine if this kind of approach is worth implementing to improve handwashing practices in various settings i.e., clinical, and educational settings.

## Limitations

The intervention, technology and experimental setup were evaluated in a pre-study. The pre-study showed promising results with regards to main effects: a statistically significant increase in handwashing time was observed in the treatment kindergarten. Minor issues and challenges that were encountered in the pre-study are considered and improved in the presented study here diminish existing limitations. Especially the technical hardware was improved to diminish the risk of hardware failures and outages. Furthermore, a research member is assigned to address the technical issues during the study immediately, if any issues happen. A possible limitation is the Hawthorne effect when observing the handwashing activity, as individuals might change their behaviour when knowing that they are being observed. To manage this, the observers will not be informed the participants when and where the observation sessions take place, and during the observations, they will be informed that the focus is on hand hygiene in general. Another limitation is the ongoing Covid-19 pandemic, as it can affect the collection of sick leave days for the children and kindergarten staff. To minimize this, sick leave data one year before the baseline phase will be obtained from the record of the kindergarten. Covid-19 also affect the children's hand hygiene behaviour and performance. Thus, there will be four different groups to be compared in three timepoints and a 12-month follow-up.

## Supporting information

**S1 Table. SPIRIT checklist.**
(PDF)

**S1 File. Ethical approval, finland (finnish original version and english translation).**
(PDF)

**S2 File. Ethical approval, germany (german original version and english translation).**
(PDF)

**S3 File.**
(PDF)

## Acknowledgments

The authors acknowledge the participating kindergartens in Turku, Finland and Bamberg, Germany for their willingness to take part in this study.

## Author Contributions

**Conceptualization:** Glenda Dangis, Joanna Graichen, Sanna Salanterä, Anni Pakarinen.

**Funding acquisition:** Sanna Salanterä, Thorsten Staake, Anni Pakarinen.

**Investigation:** Glenda Dangis, Kirsi Terho, Joanna Graichen, Sebastian A. Günther, Carlo Stingl, Anni Pakarinen.

**Methodology:** Joanna Graichen, Sebastian A. Günther, Sanna Salanterä, Thorsten Staake, Carlo Stingl.

**Project administration:** Kirsi Terho, Joanna Graichen, Riitta Rosio, Sanna Salanterä, Thorsten Staake, Anni Pakarinen.

**Supervision:** Sanna Salanterä, Thorsten Staake, Anni Pakarinen.

**Writing – original draft:** Glenda Dangis.

**Writing – review & editing:** Kirsi Terho, Joanna Graichen, Riitta Rosio, Sanna Salanterä, Carlo Stingl, Anni Pakarinen.

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
