## [Decision Letter · Decision Letter 0]

20 Sep 2022

PONE-D-22-16014Hand hygiene of kindergarten children - understanding the effect of live feedback on handwashing behaviour, self-efficacy, and motivation of young children: protocol for a multi-arm cluster randomized controlled trialPLOS ONE

Dear Dr. Dangis,

Thank you for submitting your manuscript to PLOS ONE. After careful consideration, we feel that it has merit but does not fully meet PLOS ONE’s publication criteria as it currently stands. Therefore, we invite you to submit a revised version of the manuscript that addresses the points raised during the review process.

Please revise the paper according to the comments made by the two reviewers.  

We look forward to receiving your revised manuscript.

Kind regards,

Maiken Pontoppidan

Academic Editor

PLOS ONE

Journal Requirements:

4. We note that the original protocol that you have uploaded as a Supporting Information file contains an institutional logo. As this logo is likely copyrighted, we ask that you please remove it from this file and upload an updated version upon resubmission.

Reviewers' comments:

Reviewer's Responses to Questions

**Comments to the Author**

1. Does the manuscript provide a valid rationale for the proposed study, with clearly identified and justified research questions?

Reviewer #1: Yes

Reviewer #2: Yes

2. Is the protocol technically sound and planned in a manner that will lead to a meaningful outcome and allow testing the stated hypotheses?

Reviewer #1: Partly

Reviewer #2: Yes

3. Is the methodology feasible and described in sufficient detail to allow the work to be replicable?

Reviewer #1: No

Reviewer #2: No

4. Have the authors described where all data underlying the findings will be made available when the study is complete?

Reviewer #1: Yes

Reviewer #2: No

5. Is the manuscript presented in an intelligible fashion and written in standard English?

Reviewer #1: Yes

Reviewer #2: Yes

6. Review Comments to the Author

You may also provide optional suggestions and comments to authors that they might find helpful in planning their study.

Reviewer #1: Thanks for the opportunity to review this trial protocol paper; this seems like an interesting and potential impactful trial. However, I find some aspects of the design and analysis to be quite unclear in the manuscript (and a couple of aspects I find concerning if accurate) and have outlined comments to this effect below:

1. The hypotheses section is unclear as to what comparisons are being made and in what phases of the trial (and this doesn’t appear to be explained elsewhere). I think it is important to pre-state these in a multi-arm trial, if not here, then in the analysis section. For example, for H1, I assume this is a comparison of the A and B groups pooled versus the control group to test superiority of the display intervention over nothing (control) in the intervention phase? H2 is then presumably a comparison of B versus A in the post and long term phase? I suggest making this clearer for all the hypotheses. The hypotheses section also doesn’t seem to mention the intervention C arm at all; I assume this arm is included in some of the comparisons.

2. In the eligibility criteria, it says the criteria will be “children enrolled in the kindergartens aged three to six years old, in single cases 2,5 years old can be included”. I don’t understand the latter part of that sentence, what does “in single cases” mean here? Surely either 2.5 year olds are eligible or they are not?

3. It is stated that the primary outcome is the hand hygiene behaviour of the children but then a number of measures are listed (“It will be measured from the system that will be supported by the parent survey, kindergarten staff interview, and children’s observation session. The system will measure the hand hygiene activities which includes the start time of extraction, water volume, water temperature, and end time of extraction; from these data, the following data will be derived: average flow rates, approximation of energy consumption, quality and duration of handwashing, and time stamp of soap used”). Will these all be reported as separate measures, or is there some composite measure that will be constructed (if so, how?)? I note later in the data analyses section it says “water and soaping data are combined to single handwashing procedures;” but I think needs more clarity as this only seems to cover some of the above mentioned measures.

4. Following on from the previous point, if there are multiple measures for the “primary outcome” and not composite, how are you interpreting these measures? In most trials, you would have a single primary outcome measure which is the most important measure and the headline result as to whether trial shows effectiveness of the intervention. I don’t believe this is always necessary, but I do think your primary outcome and subsequent interpretation needs to be clearer. Are you using a frequentist approach and using p-values to interpret “statistically significant” results (that is my assumption based on your sample size calculation)? If you are, how will you interpret these if some measures are statistically significant (and clinically significant), and some are not? I am not necessarily an advocate of correcting p-values for multiple comparisons, but I think there needs to be some indication of how will interpret these multiple measures as showing whether trial shows effectiveness of the intervention. Is there some hierarchy of which are most important to show benefit?

5. In the sample size section, you say “With a cluster size of 40, this means 3 clusters/kindergartens”. Do you mean 3 clusters of 40 participants per arm? This needs to be made clear. (I assume can’t be 3 clusters total, as you have 4 arms…). I also think you need to state target sample size (participants and clusters) in the abstract.

6. In the randomisation section, you say that “The participants will be allocated randomly to control, intervention A (instruction), intervention B (reward), and intervention C (reward plus instruction) groups”. I don’t understand this – surely it is the clusters that are being randomised and not the participants as you will be applying each intervention to a whole cluster/kindergarten? Please clarify.

7. There needs to be a bit more information about the randomisation algorithm to assure the reader that this process is not biased. How will the researcher randomly assign the clusters (assume it is clusters as per above comment)? Will they write/use an algorithm in some software package, use some online randomisation service/tool? Presumably they will use stratified block randomisation to ensure equal number of clusters across arms/strata?

8. The manuscript says “this study will use double-blinding in which the children, parents, kindergarten staff, and the statistician will be unaware of which group received the content of the intervention”. I don’t understand how it is possible to blind the children and the kindergarten staff, will they not be able to tell based on the installation? For example, your manuscript says that “The control group will have no screen display next to the sink throughout the study.”

9. It is not 100% clear from the data management section how the quantitative hand washing measures will be collated for purpose of data analysis; it says no person-specific data will be collected, presumably this will be anonymised individual data (which can’t be linked to child specific data) or will it be summary data from the cluster? This then affects the quantitative analysis section presumably.

10. The quantitative analysis part of the data analyses section needs more detail. Some of this is because it is not clear what measures form part of the primary outcome or which comparisons are being made (as per previous comments). The protocol doesn’t need to include the full statistical analysis plan, but at minimum I’d expect this section to include the analysis strategy for the primary outcome including the analytic model, which covariates (fixed or random effects) will be included, what groups will be compared at which timepoints and how missing data will be treated.

11. “The intervention groups are Intervention A is the instruction group; Intervention B is the reward group; and Intervention C is the instruction plus reward group”. I find this sentence misleading. The diagram you have provided seems clear, and my understanding is Intervention B does include instructions in the intervention phase but not in the post-phase. I suggest editing this sentence to reflect that.

12. I don’t necessarily agree with the argument that you don’t need a Data Monitoring Committee or other independent oversight, just because there are no perceived safety risks. This is not the only reason to have one; ensuring good study design, good clinical practice and data quality is another. Is there a Trial Steering Committee or other independent oversight? I suggest stating if there is. (If not, I recognise that this is probably not something you can change at this stage).

Reviewer #2: Thank you for the opportunity to review this protocol paper. This is a very important and valuable study understanding the effects of hand washing behaviour of kindergarten children through an intervention of live feedback, observations, and interviews/survey. I would be very interested to hear your study outcomes in the next 12 or so months! While this paper is well written, overall, some areas of the protocol need to be clearer. I have provided some comments that authors might like to consider.

Introduction:

• Paragraphs 2 to 5: suggest rearranging, i.e. start with the importance of education settings for child development (this will link to the first paragraph), then high infectious disease transmission in these educational settings (problem), then the importance of hand hygiene (solution), but how this is difficult to monitor and currently data around this are limited etc.

• Since the theme of your study is based on gamification and feedback (live feedback in particular) I think you need to include current literature on how gaming and feedback can impact change in behaviour.

Methods:

• Objective and Hypotheses

o Unclear in the study how you will measure increase self efficacy and motivation crowding-out

• Hypotheses

o Intervention C seemed to have been left out of this section – this needs to be included in your hypotheses

o Crowding-out – not familiar with this term – can you provide an example and/or more explanation?

• Trial design

o Second paragraph, Line 7, page 7 ‘… Intervention B is the reward group;…’ Doesn’t intervention B include instruction as well as reward, but Intervention C include instructions post intervention?

o Page 8, line 1 ‘… which will receive instructions but no reward.’ Interested to know authors’ thoughts on this – will children be disappointed seeing instructions but receiving no reward? Will this result in the decrease in hand hygiene?

o I’m wondering if a true control would be a group with no screens installed, just the monitor system; would having screens installed have any effects (the very least curiosity in children) on their behaviour?

• Eligibility criteria

o Not sure what ‘2,5 years old’ meant? Is that 2 to 5 years, or 2 and 5? Or do you mean 2.5?

o Unclear who is to give informed consent – children or their parents/guardians? Can a 3 or 4, even 5 yo give informed consent?

o Page 10, line 4 ‘The hand hygiene behaviour of the children will be observed…’ Who will be doing the observation? Will the kindergarten centre staff have time to observe and record their behaviour? How will be observations be recorded?

o Children interviews – can you provide the types of sample questions? These can be quite complex for a 3yo. And will it be a group discussion or a one-on-one interview? How will you engage those ‘shy’ and perhaps non cooperative children? If you ‘exclude’ them, will you be introducing bias in the study?

• Study participants

o You need to explain figure 5 in more detail or move (Fig 5) to a more relevant section. It doesn’t seem to fit currently.

• Sample Size

o You have mentioned a pilot study a couple of times throughout the manuscript; however, you haven’t provided any details. Details and how this study relates to the pilot study is needed to provide context.

o For those who struggles with sample size calculations and just pass this onto their statisticians (�), can you please, in simpler terms, tell us how many in each group i.e how many kindergartens in control/intervention groups (3 each?) and how many children needed to participate in each kindergarten? Will the numbers be similar across groups and within groups? Expected numbers at baseline, dropouts, and minimum numbers needed to provide significance in the study?

• Randomization

o Page 11, paragraph 2: It is important to be consistent when you use the term ‘participants’ – do you mean kindergarten or children? If participants refer to kindergarten, please say that, as participants are usually referred to as individuals. If this is referred to as individuals, i.e. children participating in this study, how will you allocate them randomly in each group at the same kindergarten and monitor them?

o Page 13: as the study is based on the volume or water and soap used and children are not always observed, is it possible for the children to leave the tap running or use a large amount of soap wasted onto the sink without washing their hands? Will that be a limitation?

o Is the purpose for the children to imitate the screen instruction while washing hands?

• Ethics and dissemination

o Page 18, line 3: ‘Consent forma’ should be ‘Consent forms’? �

o In the eligibility criteria section, you mentioned children giving informed consent (see comment above), but here, you say it is decided by parent or legal guardian – can you please make this clearer who is to provide consent, and how is informed consent be given by the children?

• Limitations

o Technical issues are not a limitation of your study design, but a risk that you need to mitigate, and have a backup plan.

o COVID-19 pandemic is a curve ball no one really see coming! In our study, we found that through COVID-19, children have been taught hand washing and hygiene so well, it impacted our study/game design! So COVID-19 is a positive for the children as it increased the hand hygiene knowledge, but not great for us researchers!

• Fig 1

o Your schedule doesn’t include any follow-up except deinstallation of faucets. Will you have 12 month follow up, observation and interviews? Where possible, please include any follow up to measure the effects of your study and whether it sustained the hand hygiene habit in participating children

o Shouldn’t you be including sick leave days at baseline, post interventions and follow up?

7. PLOS authors have the option to publish the peer review history of their article (what does this mean?). If published, this will include your full peer review and any attached files.

Reviewer #1: No

Reviewer #2: **Yes: **Ruby Biezen

---

## [Author Response · Author response to Decision Letter 0]

16 Nov 2022

Response to the Reviewers on PONE-D-22-16014

Dear Editor and Reviewers,

Thank you for your kind words and insightful review of our submission. We have addressed your important points, comments, and suggestions raised. The responses are documented in the table below.

Academic Editor’s comments and author’s responses:

-A rebuttal letter that responds to each point raised by the academic editor and reviewer(s). You should upload this letter as a separate file labeled 'Response to Reviewers'.

-A marked-up copy of your manuscript that highlights changes made to the original version. You should upload this as a separate file labeled 'Revised Manuscript with Track Changes'.

-An unmarked version of your revised paper without tracked changes. You should upload this as a separate file labeled 'Manuscript'.

Response: In the current revision, the rebuttal letter was uploaded as a separate file named “Response to Reviewers”. The marked-up copy of the manuscript is also uploaded as a separate file named “Revised Manuscript with Track Changes”. The unmarked version of the revised manuscript without track changes is also uploaded as separate file named “Manuscript”

2. If applicable, we recommend that you deposit your laboratory protocols in protocols.io to enhance the reproducibility of your results. Protocols.io assigns your protocol its own identifier (DOI) so that it can be cited independently in the future.

Response: Thank you for the suggestion. This study is not a laboratory protocol, but rather a study protocol of a cluster randomized controlled trial. Hence, we cannot deposit in protocol.io.

Response: The manuscript has been edited following PLOS ONE’s style requirements, thank you for the comment.

4. We note that you have stated that you will provide repository information for your data at acceptance. Should your manuscript be accepted for publication, we will hold it until you provide the relevant accession numbers or DOIs necessary to access your data. If you wish to make changes to your Data Availability statement, please describe these changes in your cover letter and we will update your Data Availability statement to reflect the information you provide

Response: Thank you for this comment. This manuscript is a study protocol, and we cannot provide repository information. However, we correct this and stated this information in the Data availability statement in the cover letter.

Response: Thank you for your comment, the full ethics statement in the method section and the full name of the ethics committees are included in the manuscript. Informed written consent from the parents and kindergarten staff will be obtained and verbal consent will be obtained from the children. These statements have been added in the ethical considerations section which can be found on pages 19 and 20.

6. We note that the original protocol that you have uploaded as a Supporting Information file contains an institutional logo. As this logo is likely copyrighted, we ask that you please remove it from this file and upload an updated version upon resubmission.

Response: This was left unnoticed, thank you for your comment. The institutional logos in the supporting information files are deleted.

Comments from Reviewers 1 and 2: 

1. Does the manuscript provide a valid rationale for the proposed study, with clearly identified and justified research questions?

Reviewer #1: Yes

Reviewer #2: Yes

Response: Thank you for the kind words, I hope that the revised manuscript is even better.

2. Is the protocol technically sound and planned in a manner that will lead to a meaningful outcome and allow testing the stated hypotheses?

Reviewer #1: Partly

Reviewer #2: Yes

Response: I acknowledge that there are sections in the manuscript which have insufficient details and some sections especially in statistical analyses should have been described clearly. The comments of the editor and the reviewers are addressed accordingly, and the responses are found below.

3. Is the methodology feasible and described in sufficient detail to allow the work to be replicable?

Reviewer #1: No

Reviewer #2: No

Response: Agreed. The methodology has insufficient details and is not described clearly. The editor’s and reviewers’ comments and suggestions are considered. In the current revision, the important points that were raised have been corrected and explained clearly.

4. Have the authors described where all data underlying the findings will be made available when the study is complete?

Reviewer #1: Yes

Reviewer #2: No

Response: Thank you. The manuscript has been edited and provided sufficient details. I hope that the current revision is better organized.

5. Is the manuscript presented in an intelligible fashion and written in standard English?

Reviewer #1: Yes

Reviewer #2: Yes

Response: Thank you, the current version has been checked and edited for any typographical or grammatical errors using grammar software.

6. Review Comments to the Author: 

Reviewer #1 comments: 

1. The hypotheses section is unclear as to what comparisons are being made and in what phases of the trial (and this doesn’t appear to be explained elsewhere). I think it is important to pre-state these in a multi-arm trial, if not here, then in the analysis section. For example, for H1, I assume this is a comparison of the A and B groups pooled versus the control group to test superiority of the display intervention over nothing (control) in the intervention phase? H2 is then presumably a comparison of B versus A in the post and long term phase? I suggest making this clearer for all the hypotheses. The hypotheses section also doesn’t seem to mention the intervention C arm at all; I assume this arm is included in some of the comparisons.

Response: Indeed, the intervention C group was missing in the hypotheses section. We also added the comparisons of the four groups (control and intervention groups A, B, and C) at different timepoints (baseline, intervention, post-intervention phase and follow-up in Finland). This section can be found on pages 6 and 7.

2. In the eligibility criteria, it says the criteria will be “children enrolled in the kindergartens aged three to six years old, in single cases 2,5 years old can be included”. I don’t understand the latter part of that sentence, what does “in single cases” mean here? Surely either 2.5 year olds are eligible or they are not?

Response: Yes, you are right this statement could have been made clearer. Children aged two years and six months old are eligible to participate in the study. This has been edited. This section can be found on page 10.

3. It is stated that the primary outcome is the hand hygiene behaviour of the children but then a number of measures are listed (“It will be measured from the system that will be supported by the parent survey, kindergarten staff interview, and children’s observation session. The system will measure the hand hygiene activities which includes the start time of extraction, water volume, water temperature, and end time of extraction; from these data, the following data will be derived: average flow rates, approximation of energy consumption, quality and duration of handwashing, and time stamp of soap used”). Will these all be reported as separate measures, or is there some composite measure that will be constructed (if so, how?)? I note later in the data analyses section it says “water and soaping data are combined to single handwashing procedures;” but I think needs more clarity as this only seems to cover some of the above mentioned measures.

Response: Thank you for the in-depth analysis of this part. The team considered your comments, and we decided that there will be only one measurement for the primary outcome which will be the data from the system. In addition, the other measurements that will be used in the secondary and tertiary outcomes will also give support to the primary outcome’s result. This section can be found on pages 10 and 11 in the outcome measures section.

4. Following on from the previous point, if there are multiple measures for the “primary outcome” and not composite, how are you interpreting these measures? In most trials, you would have a single primary outcome measure which is the most important measure and the headline result as to whether trial shows effectiveness of the intervention. I don’t believe this is always necessary, but I do think your primary outcome and subsequent interpretation needs to be clearer. Are you using a frequentist approach and using p-values to interpret “statistically significant” results (that is my assumption based on your sample size calculation)? If you are, how will you interpret these if some measures are statistically significant (and clinically significant), and some are not? I am not necessarily an advocate of correcting p-values for multiple comparisons, but I think there needs to be some indication of how will interpret these multiple measures as showing whether trial shows effectiveness of the intervention. Is there some hierarchy of which are most important to show benefit?

Response: Yes, thank you for the detailed explanation of the statistical analysis. As the team decided, we will only use one measurement for the primary outcome which will be the data from the system. There will also be five secondary outcomes: (1) Self-efficacy – children’s one-on-one interview after baseline, intervention, and post-intervention, (2) intrinsic motivation – children’s one-on-one interview after baseline, intervention, and post-intervention, (3) hand hygiene behaviour at home – parent’s survey after intervention phase, (4) hand hygiene behaviour in kindergarten based on kindergarten staff – kindergarten staff interview after intervention phase, (5) hand hygiene behaviour based on direct observation – research member’s observation sessions after baseline, intervention, and post-intervention. The tertiary outcome will be the same, sick leave absences comparing one year before and after the baseline phase. This section can be found on pages 10 and 11.

5. In the sample size section, you say “With a cluster size of 40, this means 3 clusters/kindergartens”. Do you mean 3 clusters of 40 participants per arm? This needs to be made clear. (I assume can’t be 3 clusters total, as you have 4 arms…). I also think you need to state target sample size (participants and clusters) in the abstract.

Response: We agree with you that this statement should have been made clearer. Since the sample size needed is 106 children, there will be 3 clusters with a size of 40 children, of which one cluster will be divided into two intervention arms. Hence, the number size of the two intervention arms will be 20 children. The other two groups control and one intervention group will consist of 40 children. A detailed explanation can be found on page 12. We also added the target sample size in the abstract section on page 3. 

6. In the randomisation section, you say that “The participants will be allocated randomly to control, intervention A (instruction), intervention B (reward), and intervention C (reward plus instruction) groups”. I don’t understand this – surely it is the clusters that are being randomised and not the participants as you will be applying each intervention to a whole cluster/kindergarten? Please clarify.

Response: Agreed. This statement needs to be corrected. The correct statement is that the kindergartens will be randomised to control and to three intervention groups. This section can be found on page 13.

7. There needs to be a bit more information about the randomisation algorithm to assure the reader that this process is not biased. How will the researcher randomly assign the clusters (assume it is clusters as per above comment)? Will they write/use an algorithm in some software package, use some online randomisation service/tool? Presumably they will use stratified block randomisation to ensure equal number of clusters across arms/strata?

Response: Thank you for pointing this out, we agree that we have not stated the details of the randomization process in the manuscript. The team decided to use simple randomization by drawing lots. Where each cluster will be given a number written on a small piece of paper which will be folded. The two personnel who will not be part of the team will be the ones to conduct the randomization. This section can be found on page 13 in the randomization section.

8. The manuscript says “this study will use double-blinding in which the children, parents, kindergarten staff, and the statistician will be unaware of which group received the content of the intervention”. I don’t understand how it is possible to blind the children and the kindergarten staff, will they not be able to tell based on the installation? For example, your manuscript says that “The control group will have no screen display next to the sink throughout the study.”

Response: Thank you for your comment. The statement could have been explained clearly. The study will use double-blinding however, a more detailed statement could have been added. The content of the informational letter to be dispersed to children, parents, and kindergarten staff will be only the faucet and the system that collects data. The live feedback display will not be mentioned. The children and the staff will be unaware if they are receiving the intervention or not. This section can be found on page 13 in the blinding section.

9. It is not 100% clear from the data management section how the quantitative hand washing measures will be collated for purpose of data analysis; it says no person-specific data will be collected, presumably this will be anonymised individual data (which can’t be linked to child specific data) or will it be summary data from the cluster? This then affects the quantitative analysis section presumably.

Response: This section would have been made clearer. It is correct that the collection of data will be done anonymously However, since this study will be a cluster randomized trial, the data collected will be the summary data for each arm. This statement is added and can be found on page 18. 

10. The quantitative analysis part of the data analyses section needs more detail. Some of this is because it is not clear what measures form part of the primary outcome or which comparisons are being made (as per previous comments). The protocol doesn’t need to include the full statistical analysis plan, but at minimum I’d expect this section to include the analysis strategy for the primary outcome including the analytic model, which covariates (fixed or random effects) will be included, what groups will be compared at which timepoints and how missing data will be treated

Response: As we have decided that we will use one measure for the primary outcome (data from the system). The outcomes will be compared among the groups in three timepoints which will be after the baseline, intervention, and post-intervention phases. Statements are also added regarding the analysis. This section can be found on pages 17 and 18.

11. “The intervention groups are Intervention A is the instruction group; Intervention B is the reward group; and Intervention C is the instruction plus reward group”. I find this sentence misleading. The diagram you have provided seems clear, and my understanding is Intervention B does include instructions in the intervention phase but not in the post-phase. I suggest editing this sentence to reflect that.

Response: Indeed, this statement is misleading. This was left unnoticed. The intervention B and C groups will receive instruction plus a reward in the intervention phase and during the post-intervention phase, the intervention C group will receive instructions. The statement has been corrected and it’s found on page 8 in the trial design section and page 15 in the implementation section.

12. I don’t necessarily agree with the argument that you don’t need a Data Monitoring Committee or other independent oversight, just because there are no perceived safety risks. This is not the only reason to have one; ensuring good study design, good clinical practice and data quality is another. Is there a Trial Steering Committee or other independent oversight? I suggest stating if there is. (If not, I recognise that this is probably not something you can change at this stage).

Response: Yes, we agree with you, this part should be explained clearer. This study will not have Data Monitoring Committee or Trial Steering Committee. This behavioural intervention poses a very low risk, it does not require trial steering or data monitoring committee (Sydes 2004). This section can be found on page 19.

Reviewer #2 comments: 

1. Introduction:

• Paragraphs 2 to 5: suggest rearranging, i.e. start with the importance of education settings for child development (this will link to the first paragraph), then high infectious disease transmission in these educational settings (problem), then the importance of hand hygiene (solution), but how this is difficult to monitor and currently data around this are limited etc.

Response: Thank you for your comments and suggestions. We have made changes according to your suggestions and added statements about the importance of hand hygiene. This can be found on pages 4 – 6. 

Since the theme of your study is based on gamification and feedback (live feedback in particular) I think you need to include current literature on how gaming and feedback can impact change in behaviour.

Response: We agree with you that gamification and feedback should be added in the introduction part. We have added one paragraph regarding this. This can be found on page 5 in the introduction section.

2. Methods:

• Objective and Hypotheses

o Unclear in the study how you will measure increase self efficacy and motivation crowding-out

Response: Thank you for noticing this. Statements are added for further details on how to measure the self-efficacy level and motivation crowding out. The measurements for these will be children’s one-on-one interviews done by the assistant researchers. The children will answer by pointing out their answer from the three smiley faces (sad, neutral, and happy) then after will be followed by open-ended questions. This can be found in the outcome measures section on pages 10 and 11.

3. Hypotheses

o Intervention C seemed to have been left out of this section – this needs to be included in your hypotheses

Response: This part was left unnoticed, thank you for the comment. Intervention C was included in the hypotheses section which can be found on pages 6 and 7.

4. Crowding-out – not familiar with this term – can you provide an example and/or more explanation?

Response: Thank you for the comment. According to Gagner 2014, crowding out is an effect where there is a decrease or lack of motivation to perform a task because the reward is already known or expected. A brief statement is added in the objectives and hypotheses section on page 6.

5. Trial design

o Second paragraph, Line 7, page 7 ‘… Intervention B is the reward group;…’ Doesn’t intervention B include instruction as well as reward, but Intervention C include instructions post intervention?

Response: Agreed, this was left unnoticed. It is correct that interventions B and C included instruction plus reward in the intervention phase and during the post-intervention phase intervention C includes instructions. This part has been corrected and it can be found on pages 8 and 9.

6. Page 8, line 1 ‘… which will receive instructions but no reward.’ Interested to know authors’ thoughts on this – will children be disappointed seeing instructions but receiving no reward? Will this result in the decrease in hand hygiene?

Response: Thank you, you made a good point on this. This is one part we wanted to investigate in this study if receiving instructions plus a reward is more effective in hand hygiene than receiving only instructions. Also, in the informational letters, the feedback display will not be mentioned, and the participants will not know about it until the intervention phase. This statement is added on page 13 in the blinding section.

7. I’m wondering if a true control would be a group with no screens installed, just the monitor system; would having screens installed have any effects (the very least curiosity in children) on their behaviour?

Response: A very good point on this but since the feedback display is not mentioned in the informational letter then the children in the control group would not have known that there will be a real-time feedback display. This statement is added in the blinding section in page 13. Installing the display without any activity in the control group might affect the children’s behaviour, but unfortunately this will not be investigated in this study. 

8. Eligibility criteria

o Not sure what ‘2,5 years old’ meant? Is that 2 to 5 years, or 2 and 5? Or do you mean 2.5?

Response: You are right this statement could have been made clearer. So, children aged two years and six months old are eligible to participate in the study. This has been edited. This section can be found on page 10.

9. Unclear who is to give informed consent – children or their parents/guardians? Can a 3 or 4, even 5 yo give informed consent?

Response: We agree that this section could have been explained more clearly. The study will be explained to the children in a manner that they can understand, and they will be asked if they want to participate in the study. Children will provide verbal consent if they want to participate or not, but it is the parent’s decision if they want their children to participate in the study or not. This statement can be found on page 10 in the eligibility criteria section and pages 19 and 20 in the ethical considerations section.

10. Page 10, line 4 ‘The hand hygiene behaviour of the children will be observed…’ Who will be doing the observation? Will the kindergarten centre staff have time to observe and record their behaviour? How will be observations be recorded?

Response: Thank you for asking about this part. We acknowledge that we have not provided complete details about the observation. Assistant researchers will be hired to interact with the children, parents, and kindergarten staff. They will do the observations at the last week of each phase. And due to the limited resources, video recording cannot be realized. This section can be found on pages 10 and 11 in the outcome measures section.

11. Children interviews – can you provide the types of sample questions? These can be quite complex for a 3yo. And will it be a group discussion or a one-on-one interview? How will you engage those ‘shy’ and perhaps non cooperative children? If you ‘exclude’ them, will you be introducing bias in the study?

Response: Thank you for having an interest in the children’s questions. It will be a one-on-one interview; we also have provided further details in the manuscript. These interview questions will be short, and easy to understand, and questions will be pre-tested. During the interview, the interviewer will introduce herself/himself first and ask simple questions regarding their day or regular activities. Simple instructions and a sample question will be given for the children to feel comfortable. This part can be found on pages 10 and 11.

●Sample questions: (1) Do you think that you can wash your hands properly? (2) Do you like washing your hands? (3) Do you feel like it is easy to wash your hands?

For shy children, assistant researchers will try their best to let them participate in the study. But in case there will be non-cooperative children, they can be excluded. In these types of cases, researchers cannot intervene more, hence we acknowledge that bias will exist.

12. Study participants

o You need to explain figure 5 in more detail or move (Fig 5) to a more relevant section. It doesn’t seem to fit currently.

Response: Indeed. Figure 5 has been moved to a more relevant section which can be found on page 15 in the implementation section. The figure is updated to figure 8, a flowchart of the cluster randomization.

13. Sample Size

o You have mentioned a pilot study a couple of times throughout the manuscript; however, you haven’t provided any details. Details and how this study relates to the pilot study is needed to provide context.

Response: Agreed. Thank you for your comment. This was corrected to a pre-study. A brief statement has been added regarding the pre-study and can be found on page 12 in the sample size section and pages 21, and 22 in the limitations section.

14. For those who struggles with sample size calculations and just pass this onto their statisticians (�), can you please, in simpler terms, tell us how many in each group i.e how many kindergartens in control/intervention groups (3 each?) and how many children needed to participate in each kindergarten? Will the numbers be similar across groups and within groups? Expected numbers at baseline, dropouts, and minimum numbers needed to provide significance in the study?

Response: Thank you for your comment. Since we based the sample size calculation on a pre-study, we must provide statistical figures. But we agree that this could have been explained in a clearer way. The calculated sample size will be 106 participants so there will be total of 3 clusters with a size of 40 children, of which one cluster will be divided into two intervention arms (Intervention B and C). These two interventions will be having instructions plus a reward during the intervention phase but during the post-intervention phase, intervention C will remain to have the instructions. Hence, the number size of the two intervention arms will be 20 children. This can be found on page 12.

.

15. Randomization

o Page 11, paragraph 2: It is important to be consistent when you use the term ‘participants’ – do you mean kindergarten or children? If participants refer to kindergarten, please say that, as participants are usually referred to as individuals. If this is referred to as individuals, i.e. children participating in this study, how will you allocate them randomly in each group at the same kindergarten and monitor them?

Response: Thank you for pointing this out, it really is important to be consistent with the term use. This statement has been corrected. The kindergartens will be randomized to control and intervention groups. This can be found on page 13.

16. Page 13: as the study is based on the volume or water and soap used and children are not always observed, is it possible for the children to leave the tap running or use a large amount of soap wasted onto the sink without washing their hands? Will that be a limitation?

Response: This is an interesting question. The faucets and soap dispensers are installed with sensors. The children can play and use a large amount of water or soap wasted onto the sink without washing their hands, but this can happen with a very small chance. Sinks are available in most kindergarten rooms where they can wash their hands aside from the toilets. In the toilets, kindergarten staff (preschool assistants) usually are on watch to help the children when needed. 

17. Is the purpose for the children to imitate the screen instruction while washing hands?

Response: Thank you for your comment. Yes, the purpose is for the children to imitate the handwashing instructions during the handwashing activity. This statement is included and can be found on pages 13 and 14 in the intervention development section.

18. Ethics and dissemination

o Page 18, line 3: ‘Consent forma’ should be ‘Consent forms’?

Response: Thank you for noticing it. The typo error has been corrected. 

19. In the eligibility criteria section, you mentioned children giving informed consent (see comment above), but here, you say it is decided by parent or legal guardian – can you please make this clearer who is to provide consent, and how is informed consent be given by the children?

Response: Thank you for your comment. We made corrections on this part to make it clearer. The children will be informed and explained about the upcoming study in an understandable manner. They will provide verbal consent to participate, however, it is the parents who will decide their participation. This part can be found on pages 10 in the eligibility criteria and pages 19 and 20 in the ethical considerations section.

20. Limitations

o Technical issues are not a limitation of your study design, but a risk that you need to mitigate, and have a backup plan.

Response: Thank you for correcting this statement, we considered your comment. We corrected and added statements regarding this. The backup plan for these technical issues would be to have spares of the hardware and a research member be assigned to address the technical issues whenever needed. This part can be found on pages 21 and 22. 

21. COVID-19 pandemic is a curve ball no one really see coming! In our study, we found that through COVID-19, children have been taught hand washing and hygiene so well, it impacted our study/game design! So COVID-19 is a positive for the children as it increased the hand hygiene knowledge, but not great for us researchers!

Response: This is exactly true. This can be a limitation, but it is not always the case for very young children, they maybe have been taught handwashing and have the knowledge of why to wash their hands, but they may still not know the proper way to wash their hands.

22. Fig 1

o Your schedule doesn’t include any follow-up except the deinstallation of faucets. Will you have 12 month follow up, observation and interviews? Where possible, please include any follow up to measure the effects of your study and whether it sustained the hand hygiene habit in participating children

Response: Thank you for your suggestions. Including a 12-month follow-up for observation and interviews would be very helpful in our study. However, due to limited resources (financially), no other follow-up measurements will be conducted aside from the collection of sick leave absences. 

23. Shouldn’t you be including sick leave days at baseline, post interventions and follow up?

Response: Thank you for your comment. Statement is added, the sick leave days are collected one year before and after the start of the baseline phase. This statement has been edited to be clearer which can be found on page 18 in the data collection part.

---

## [Decision Letter · Decision Letter 1]

6 Jan 2023

Hand hygiene of kindergarten children - understanding the effect of live feedback on handwashing behaviour, self-efficacy, and motivation of young children: protocol for a multi-arm cluster randomized controlled trial

PONE-D-22-16014R1

Dear Dr. Dangis,

We’re pleased to inform you that your manuscript has been judged scientifically suitable for publication and will be formally accepted for publication once it meets all outstanding technical requirements.

Kind regards,

Maiken Pontoppidan

Academic Editor

PLOS ONE

Additional Editor Comments (optional):

Congratulations! The paper is much improved and is now ready for publication. 

Reviewers' comments:

Reviewer's Responses to Questions

**Comments to the Author**

1. Does the manuscript provide a valid rationale for the proposed study, with clearly identified and justified research questions?

Reviewer #1: Yes

Reviewer #2: Yes

2. Is the protocol technically sound and planned in a manner that will lead to a meaningful outcome and allow testing the stated hypotheses?

Reviewer #1: Yes

Reviewer #2: Yes

3. Is the methodology feasible and described in sufficient detail to allow the work to be replicable?

Reviewer #1: Yes

Reviewer #2: Yes

4. Have the authors described where all data underlying the findings will be made available when the study is complete?

Reviewer #1: Yes

Reviewer #2: Yes

5. Is the manuscript presented in an intelligible fashion and written in standard English?

Reviewer #1: Yes

Reviewer #2: Yes

6. Review Comments to the Author

You may also provide optional suggestions and comments to authors that they might find helpful in planning their study.

Reviewer #1: Thanks to the authors for addressing my concerns and making clarifications in the manuscript. I am happy that all of my comments have been addressed and I have no further comments.

Reviewer #2: Thank you for the opportunity to review this manuscript revision. As mentioned in the first review, this is a very important study; the authors have now done a great job addressing the reviewers comments. I am satisfied that my concerns have been addressed appropriately.

My only comment/feedback is Figure 2, the design and data collection of the study. The figure is set out like a timeline, but of course, this is not proportional. It will be good to see the study designed displayed in another way or in a proportional timeline if possible.

Good luck with your study and would love to read the outcome when available!

7. PLOS authors have the option to publish the peer review history of their article (what does this mean?). If published, this will include your full peer review and any attached files.

Reviewer #1: No

Reviewer #2: **Yes: **Ruby Biezen

---

## [Editor Report · Acceptance letter]

16 Jan 2023

PONE-D-22-16014R1 

Hand hygiene of kindergarten children - understanding the effect of live feedback on handwashing behaviour, self-efficacy, and motivation of young children: protocol for a multi-arm cluster randomized controlled trial 

Dear Dr. Dangis:

I'm pleased to inform you that your manuscript has been deemed suitable for publication in PLOS ONE. Congratulations! Your manuscript is now with our production department. 

Kind regards, 

on behalf of

Dr. Maiken Pontoppidan 

Academic Editor

PLOS ONE